# Rheological Properties and Setting Kinetics of Bioceramic Hydraulic Cements: ProRoot MTA versus RS+

**DOI:** 10.3390/ma16083174

**Published:** 2023-04-18

**Authors:** Arne Peter Jevnikar, Tine Malgaj, Kristian Radan, Ipeknaz Özden, Monika Kušter, Andraž Kocjan

**Affiliations:** 1Endodent d.o.o., Metelkova ulica 15, 1000 Ljubljana, Slovenia; 2Department of Prosthodontics, Faculty of Medicine, University of Ljubljana, Hrvatski trg 6, 1000 Ljubljana, Slovenia; 3Department of Inorganic Chemistry and Technology, Jožef Stefan Institute, Jamova 39, 1000 Ljubljana, Slovenia; 4Department for Nanostructured Materials, Jožef Stefan Institute, Jamova 39, 1000 Ljubljana, Slovenia; 5Jožef Stefan International Postgraduate School, Jamova 39, 1000 Ljubljana, Slovenia

**Keywords:** hydraulic calcium silicate cements (HCSCs), mineral trioxide aggregate (MTA), bioactive materials, handling, rheological properties, setting kinetics

## Abstract

Hydraulic calcium silicate-based cements (HCSCs) have become a superior bioceramic alternative to epoxy-based root canal sealers in endodontics. A new generation of purified HCSCs formulations has emerged to address the several drawbacks of original Portland-based mineral trioxide aggregate (MTA). This study was designed to assess the physio-chemical properties of a ProRoot MTA and compare it with newly formulated RS+, a synthetic HCSC, by advanced characterisation techniques that allow for in situ analyses. Visco-elastic behaviour was monitored with rheometry, while phase transformation kinetics were followed by X-ray diffraction (XRD), attenuated total reflectance Fourier transform infrared (ATR-FTIR), and Raman spectroscopies. Scanning electron microscopy with energy-dispersive spectroscopy, SEM-EDS, and laser-diffraction analyses was performed to evaluate the compositional and morphological characteristics of both cements. While the kinetics of surface hydration of both powders, when mixed with water, were comparable, an order of magnitude finer particle size distribution of RS+ coupled with the modified biocompatible formulation proved pivotal in its ability to exert predictable viscous flow during working time, and it was more than two times faster in viscoelastic-to-elastic transition, reflecting improved handling and setting behaviour. Finally, RS+ could be completely transformed into hydration products, i.e., calcium silicate hydrate and calcium hydroxide, within 48 h, while hydration products were not yet detected by XRD in ProRoot MTA and were obviously bound to particulate surface in a thin film. Because of the favourable rheological and faster setting kinetics, synthetic, finer-grained HCSCs, such as RS+, represent a viable option as an alternative to conventional MTA-based HCSCs for endodontic treatments.

## 1. Introduction

In the past decade, hydraulic calcium silicate-based cements (HCSCs) have become good alternatives to epoxy-based root canal sealers in endodontics. This group of materials has been shown to possess good bioactivity, which enhances biological healing processes [1]. In addition to sealing ability, hydraulic cements are the material of choice in various clinical indications, including closure of open apices [2], retrograde root canal obturation in apical surgery, direct/indirect pulp capping [3], and repair of internal and external root resorptions [4,5]. Moreover, due to their proven biological properties, hydraulic cements are successfully used in the repair of the furcal perforations, in which it is difficult to control moisture content [6]. In addition to favourable biological and physical properties, clinicians require easy handling characteristics of the material in everyday routine. One of the first bioactive cements used clinically for more than two decades is mineral trioxide aggregate (MTA), developed by Torabinejad et al. [7]. MTA has been shown to induce mineralisation beneath exposed pulp and was initially used for root perforations and as a root end filling material [8,9]. Due to evidence-based clinical success [10,11,12,13], the indications for the use of MTA have grown considerably. Until recently, a variety of MTA-based materials was introduced into clinical practice]. However, the ideal endodontic repair material should be dimensionally stable, non-resorbable, radiopaque, biocompatible, and bioactive, and it should have good handling properties [1].

MTA is a Portland cement-based endodontic material made originally by combining grey Portland cement with a radiopaque, brownish-coloured bismuth oxide (Bi_2_O_3_) powder. The principal compounds of MTA are tricalcium silicate (C3S—Ca_3_SiO_5_) and dicalcium silicate (C2S—Ca_2_SiO_4_), but MTA also consists of considerable amounts of other oxides, such as tricalcium aluminate, calcium sulphate, tricalcium oxide, and iron oxide, which are responsible for final physio-chemical properties [14,15]. The original MTA was grey and often caused tooth discoloration. In 2002, “tooth-coloured” white MTA was introduced. Both materials have similar compositions; however, white MTA was reported to have fewer iron oxide traces and a finer particle size [16].

Despite evidence-based clinical success [10,11,12,13], several limitations have been reported with MTA, including poor handling characteristics, long setting time [17,18,19], difficult retrieval from the operation area [20], and post-treatment tooth discoloration [21,22]. Since it is based on Portland cement, an additional concern with respect to clinical employment was in its purity regarding the traces of heavy metals present in MTA-based materials, such as arsenic, bismuth, cadmium, chromium, copper, iron, lead, manganese, nickel, and zinc [23,24,25]. The sandy nature, resulting in a lack of uniformity of the premixed MTA paste, has caused some clinical limitations in paste application. Since the introduction of MTA, one of its greatest drawbacks has been the long setting time. It has been shown that MTA sets slowly in approximate 3–4 h [16,26,27]. Clinicians require a material that would ideally set before the end of the procedure.

In light of the above-mentioned drawbacks, original MTA-type formulations have been modified, offering new, purified (and/or synthetic) HCSCs systems, including Biodentine, Bioaggregate, Endosequence, and others [1,28,29]. The common material evolution was in the simplification of the compositions, while also increasing their biocompatibility. For example, Bi_2_O_3_ was replaced with biomedical grade zirconia (3Y-TZP), while the active ingredient C3S (with or without C2S) was coupled with calcium carbonates, silicates, or phosphates to promote setting and/or remineralisation [27,30]. Recently, RS+ bioceramic root canal repair material, a product from Jožef Stefan Institute’s spin-off company, has been introduced to the dental community, claiming improved handling and biological properties compared to the original MTA formulations. It is based on synthetic C3S and zirconia as a radioopacifier, with small additions of biocompatible phyllosilicate clay (bentonite) and bioactive amorphous calcium silicate for the enhanced handling, setting, and remineralisation properties of powder-like materials. The bioactive amorphous calcium silicate was shown to be a superior osteoinductive compared to calcium phosphates, such as beta-calcium phosphate and hydroxyapatite, which are commonly added to similar formulations [31,32].

Clinically, the setting time of cement paste is defined by the transition time from a fluid state into a solidified state [33]. Setting time can be affected by mixing method, quantity of water used, packaging force, and moisture of the environment. In addition to the needle penetrations using Gilmore weights, as defined by the ISO 6876 and ISO 9917-1 standards to determine setting time [34], there are several other means to experimentally evaluate setting time, including microhardness and strength measurements [35]. These methods provide some information about the setting characteristics; however, the setting reaction is observed indirectly. Moreover, handling, another very important property of successful clinical application, has rarely been tested and defined.

Handling properties and viscoelastic-to-elastic transition during setting can be precisely measured by rheometry in rotational and oscillatory modes, respectively. In situ X-ray diffraction, Fourier transform infrared (FTIR), and Raman spectroscopies can continuously follow the setting mechanism of hydraulic cements, which can in turn estimate the amounts of the constituents in samples and the transition kinetics. The purpose of this study was to evaluate the physio-chemical properties, in terms of handling and setting behaviours, of a benchmark MTA formulation ProRoot MTA hydraulic cement (Dentsply Tulsa Dental, Tulsa, OK, USA), and to compare them with those of newly formulated RS+ (Genuine Technologies d.o.o.; Ljubljana, Slovenia), by advanced characterisation techniques that allow for in situ observations of the phase and visco-elastic transformation kinetics, such as X-ray diffraction (XRD), rheometry, FTIR, and Raman spectroscopies. In addition, scanning electron microscopy with energy-dispersive spectroscopy (SEM-EDS) and laser-diffraction analyses were performed to evaluate the compositional and morphological characteristics of both cements affecting the physio-chemical properties. The null hypothesis was that there are no differences in compositional or morphological characteristics, in hydration and phase transformation kinetics, and in rheological behaviour between ProRoot MTA and a novel RS+.

## 2. Materials and Methods

### 2.1. Materials and Sample Preparation

The two hydraulic cements analysed in this research (ProRoot MTA and RS+) were prepared according to the manufacturers’ instructions. When mixing the ProRoot MTA, a new pouch of material was opened and dispensed on a mixing pad, followed by a ProRoot MTA liquid ampule being opened and its contents being squeezed out next to the material. The powder and the liquid were gradually incorporated using a mixing spatula until a thick, creamy consistency was formed. In the case of RS+, 0.3 g of powder was mixed with six drops of deionised water. The powder was first weighed on an electronic scale and then transferred to a mixing plate. Drops of distilled water were placed, and next, the powder and the liquid were incorporated using a mixing spatula until the desired consistency was obtained. As-prepared paste was then transferred to the given measuring cell (XRD, FTIR). Measurements were performed at room temperature. In the case of hydration experiments, the paste consistency of cement immediately after mixing were allowed to set over 48 h at 37.5 °C and saturated under humidity in an incubator. Afterwards, the hydrated cement was crushed in a mortar and analysed with XRD. The experimental flowchart is presented in Figure 1.

### 2.2. Scanning Electron Microscopy Coupled with Energy-Dispersive Spectroscopy (SEM-EDS)

As-received powders were analysed by SEM-EDS to evaluate their particulate morphology and elemental composition. SEM analyses were performed using an FEI Helios NanoLab focused ion beam (FIB)-SEM (Helios Nanolab 650, FEI, Hillsboro, OR, USA). Measurements of chemical composition were performed with an 50-mm^2^ X-max silicon drift detector (SDD) (Oxford Instruments, Abingdon, UK) attached to the microscope. Prior to the analysis, the powders were mounted on a carbon tape and sputtered with carbon. The SEM imaging was performed at an accelerating voltage and a beam current of 5 kV and 0.1 nA, respectively. EDS mapping was performed at a 15-kV accelerating voltage and 10-μs dwell time to obtain qualitative visualisation of the distribution of elements greater than the detection limit within the particulate samples.

### 2.3. Particle Size Distribution

The particle size distributions (PSDs) of both powders and their mixtures were determined using the laser diffraction method (Horiba LA-920, Kyoto, Japan). To avoid hydration reactions, 2-propanol was used as a solvent. Prior to the measurements, the non-aqueous powder dispersions were rigorously mixed and ultrasonicated for 5 min in the measuring cell. Volume and cumulative volume particle distributions were plotted.

### 2.4. XRD Phase Characterisation

The X-ray diffraction (XRD) data from (un)hydrated powders were collected with a Malvern Panalytical Empyrean X-ray diffractometer (Empyrean multipurpose X-ray diffractometer, Almelo, The Netherlands) using a monochromated X-ray beam produced by a Cu-target tube (λKα1 = 0.15406 nm and λKα2 = 0.154439 nm). The measurements were obtained in Bragg-Brentano geometry using a range of 10° < 2Θ < 80°, a step size of 0.0131°, a divergence slit of 0.04 rad, and a counting time of 1 s per step. In addition, the in situ hydration measurements were performed on Bragg-Brentano geometre in a range of 41° < 2Θ < 42°, applying a step size of 0.0131° and using a divergence slit of 0.04 rad and a counting time of 1 s per step. On the diffracted part, the large β-Ni filter was used to reduce the intensity of the Kα2 line. After mixing of the powder with the deionised water to produce a suitable paste, the latter was spread onto the in situ holder and measured at different times. In situ experiments collected data at different times at 2 min, 5 min, 10 min, 15 min, 20 min, 25 min, 30 min, 40 min, 1 h, and 2 h. The XRD data were analysed using HighScore Plus XRD Analysis Software database PDF-4+.

### 2.5. Rheology

Rheological characterisation of the cement pastes was performed in rotational, as well as oscillatory tests, using a Physica MCR 301 rheometer (Anton Paar GmbH, Graz, Austria). In rotational mode, the change in the viscosity of cement pastes was measured for both samples at a constant temperature of 25 °C and a constant shear rate of 10 s^−1^ for 20 min with a plate-on-plate system with a 15-mm upper plate. The setting kinetics were monitored by oscillatory measurements following the real-time build-up of the storage modulus G′ (Pa) under a sinusoidal strain of 0.0005% with an oscillation frequency of 1 radian per second. During the measurements, the plate-on-plate system with a 15-mm upper plate was used, and a gap of 0.051 mm was set between the two plates. A constant temperature of 25 °C was maintained for the duration of the measurements.

### 2.6. Infrared Spectroscopy

The measurements were performed using a PerkinElmer Spectrum 100 Fourier transform infrared (FT-IR) spectrometer (PerkinElmer, Waltham, MA, USA) equipped with a universal attenuated total reflectance (UATR) module. After an air background calibration, the freshly mixed cement pastes were placed directly on the optic window of the diamond ATR top plate fitted with a metal sample holder (powder cup). Each paste was loaded in the sample compartment, where it was closed and compressed by the pressure applicator tip with enough pressure to ensure good and uniform contact between the diamond and the sample. The spectra were collected at different times (soon after mixing and at 1.5 min, 4 min, 6 min, 8 min, 10 min, 12 min, 14 min, 20 min, 25 min, 30 min, 40 min, 45 min, 50 min, 1 h, 1.5 h, 2 h, 3 h, 4 h, 5 h, 7 h, 10 h, 72 h, and 76 h after mixing) in the 300–4000 cm^−1^ range, with a spectral resolution of 4 cm^−1^ as average spectra out of 16 scans. The spectrometer was running Spectrum 10 software (PerkinElmer, Waltham, MA, USA).

### 2.7. Raman Spectroscopy

Spectra were collected at room temperature using a Horiba Jobin-Yvon LabRam HR confocal Raman system, coupled with an Olympus BXFM-ILHS microscope. Samples were excited by the 633-nm emission line of a He−Ne laser with a power output of 7 mW on the sample through a 10× microscope objective. Thirty scans per sample were averaged over the spectral range of 100–4000 cm^−1^ with a spectral resolution of 2 cm^−1^ and exposure time of 1 s. The cement pastes were prepared directly on the microscope glass slides and measured at the same time intervals as for the FT-IR spectroscopic measurements.

## 3. Results

### 3.1. Morphological and Particle Size Analyses

As-received powders were inspected with SEM to observe their respective particulate morphologies (Figure 2). As seen on a lower-magnification SEM micrograph (Figure 2a) ProRoot MTA powder is composed of irregularly shaped particles with a broad size distribution ranging from one to several tens of micrometres in size. A higher-magnification SEM micrograph (Figure 2a) revealed that agglomerated clusters of finer particles of irregular shapes were attached to the larger particles. Particles resembled sharp edges and vertices. RS+ powder, on the other hand, was substantially finer as seen from the lower-magnification SEM micrograph (Figure 2b). It was only possible to assess the particulate morphology from the higher magnification micrograph, on which it can be seen that larger, micron-sized cuboid particles were covered with finer, 100 nm-sized spherical particles. The larger particles correspond to C3S, while the smaller match the size of zirconia (3Y-TZP) and amorphous bioactive calcium silicate.

Laser diffraction was used to analyse the particle size distributions (PSDs) of both cements (Figure 3). ProRoot MTA had a broader monomodal PSD with mean particle size (d_mean_) of approx. 15 µm, in accordance with SEM (Figure 2a). RS+ had a narrower monomodal PSD distribution with an order of magnitude-finer d_mean_ value of 1 µm. The broadness of the PSD can be depicted by d_10_, d_50_, and d_90_ from cumulative volume distributions. ProRoot MTA exhibited d_10_, d_50_, and d_90_ of 3.2, 11.0, and 28.2 µm, respectively. RS+ resembled d_10_, d_50_, and d_90_ of 0.5, 0.9, and 1.9 µm, respectively. The broadness of the PSD is reflected in the magnitude between d_10_ and d_90_. In the case of ProRoot MTA, d_90_ was more than seven times higher than d_10_, while in the case of RS+, this increase was four fold.

### 3.2. Elemental Composition

SEM-EDS mapping analysis of both cements was performed to observe the elemental distribution (Figure 4). The major elements in both cements are, as expected, calcium and silicon due to C3S being the major phase and, in the case of ProRoot MTA, owing to the C2S presence as well. In ProRoot MTA, large, micrometre-sized regions of bismuth are visible, indicating the size of Bi_2_O_3_ particulate inclusions as radioopacifying agents. In the case of RS+, zirconia is used and is distributed homogeneously throughout the sample owing to the much smaller particles (Figure 2b). ProRoot MTA contained several other elements at greater the detection limit, i.e., sulphur, aluminium, and iron, corresponding to calcium sulphate, tricalcium aluminate, and iron oxide, respectively. It is interesting to note the variation in the concentration of calcium in RS+.

### 3.3. XRD Phase Composition

Unhydrated and 48-h hydrated cement powders were analysed by XRD. In Figure 5a, the diffractogram of as-received ProRoot MTA indicates the presence of several main phases. These phases are bismuth oxide, tricalcium silicate, dicalcium silicate, and calcium sulphate, in accordance with the EDS-SEM analysis (Figure 4). Calcium aluminate was not detected, indicating its presence in traces, i.e., less than a few wt.%. Surprisingly, after hydration of ProRoot MTA, no visible changes in phase composition were detected, except for the change in the relative intensities of the peaks. In the case of RS+ powder, the XRD diffractograms before and after hydration are presented in Figure 6. As-received powder (Figure 6a) was only composed of zirconia (tetragonal and monoclinic polymorphs) and C3S. Hydrated RS+ powder yielded more significant changes compared to the case of ProRoot MTA. The C3S was completely replaced with the new phases, i.e., portlandite (Ca(OH)_2_) and calcium silicate hydrate (CSH) as main hydration products, indicating the completed hydration reaction in 48 h. In addition, traces of calcium carbonate were also detected. Compared to ProRoot MTA, the diffractogram of the as-received RS+ powder indicated the finer crystallinity and the presence of amorphous phases (due to the addition of bioactive calcium silicate) due to broader diffraction peaks and the higher background, respectively.

The higher reactivity of the RS+ powder allowed for in situ XRD measurements of the hydration kinetics in the 41–41.5° 2θ range (Figure 6b). The hydration reaction was monitored by observing (012) the peak of C3S positioned at 41.148° 2θ (PDF 00-031-0301). While there were no significant changes in the intensity of the (peak) after 5–40 min of hydration in the XRD setup, the diffraction peak was gone after 120 min, indicating the depletion of the phase with the hydration reaction.

### 3.4. FTIR and Raman Spectroscopies

FTIR was employed to monitor the surface changes in the cements during the hydration process (Figure 7 and Figure 8). Due to the strong overlap of the broad band of H_2_O vibration and absence of clear time-dependent changes less than 1000 cm^−1^, the 1300–1800 cm^−1^ and 2600–3800 cm^−1^ regions were selected and expanded for the study of the early stages of cement hydration. Both hydraulic cements, ProRoot MTA (Figure 7), and RS+ (Figure 8), exhibited similar spectral evolution during the hydration process, as evidenced by the decrease in the intensities of the characteristic broad water O–H stretching absorption in the range of 2800–3700 cm^−1^ and of the sharper H–O–H bending mode at approximately 1640 cm^−1^ [36]. The change in the aforementioned bands was minimal in the first 14 min following the mixing with mixing solution or deionised water, but a rapid decrease in the intensities can be observed at up to 40 min, after which the associated spectra are virtually identical.

In both analysed materials, ProRoot MTA and RS+, the decrease in intensity of the water bands is paralleled by the appearance of a band centred around 3640 cm^−1^, which can be ascribed to the O–H stretching mode of Ca(OH)_2_ (portlandite). At the onset of the hydration reaction, this band appears as a weak shoulder protruding from the broad H_2_O region, and over time, its intensity increases while becoming better resolved.

The time-dependent Raman spectra of ProRoot MTA and RS+ aqueous cement pastes in the 50–4000 cm^−1^ range are shown in Figure 9. ProRoot spectra exhibit sharp bands with high intensities in the 50–1000 cm^−1^ region, where the majority of all vibration modes are present. An interesting feature of ProRoot MTA is also the absence of O–H stretching vibrations of water molecules in the 3000–3800 cm^−1^ region, while a broad corresponding band can be clearly observed in RS+ soon after mixing. Due to the high spatial resolution of the method, the Raman measurements were repeated on a second RS+ paste to assess the average response of the sample, and the general spectral features were found to be highly comparable at all time intervals. The most prominent time-dependent change in the RS+ sample is represented by the band at 1083 cm^−1^, which arises soon after mixing and becomes the dominant signal approximately after 8 min following the mixing. This band can be assigned to the formation of different phases of CaCO_3_, namely to the symmetric (ν_1_) stretching of the C–O bonds, and it indicates a rapid uptake of CO_2_ upon exposure of the pasters to air [37]. Traces of CaCO_3_ were also detected with XRD. No evidence of the carbonation can be observed in ProRoot MTA, where the appearance of the band at 1008 cm^−1^ might be attributed to the symmetrical stretching of Q^2^ silicate units (chains), ν_1_(SiO_4_) [38], which appears soon after mixing but does not grow in intensity over time. Both the FTIR and Raman spectra of ProRoot MTA obtained in this study are similar to those reported by other research groups [39,40,41], with differences in the degrees of cement hydration and atmospheric carbonation due to different sample preparations and measuring setups used.

### 3.5. Rheological Properties

Rheological property measurements of the cement pastes immediately after hydration were performed in rotational, as well as in oscillatory, rheometry to assess viscosities and viscoelastic-to-elastic transition, respectively. In rotational mode (Figure 10a), the change in the viscosity (at a constant shear rate) with time provides a direct indication of the setting hydration reactions. In the case of RS+ aqueous pastes, the initial viscosity decreased from 911.8 to 473.3 Pa∙s in the first 2–3 min During this “working” period, it exhibited a shear-thinning behaviour typical for thick ceramic suspensions [42]. Afterwards, the viscosity started to steadily increase, becoming shear thickening and indicating the hydration setting reactions. After 17.5 min, soon after reaching the highest viscosity value of 24,050 Pa∙s, a sudden drop occurred that corresponded to the sample solidification and (partial) detachment from the measuring cell. On the other hand, it was not possible to obtain a legitimate flow curve behaviour for ProRoot MTA paste. The repetitive increase and decrease in the viscosity were due to the inhomogeneous, sandy-like paste state, which was unable to exhibit a viscous fluid-like flow. The reason for this outcome might be a relatively large PSD (Figure 3) and/or partial solidification of the material.

The oscillatory mode was used to observe the viscoelastic properties of both pastes during setting and to observe viscoelastic-elastic transition (Figure 10b). The material is initially viscoelastic, in which the interlocking of cement begins to occur on account of the precipitation of crystalline products. When this process occurs, the storage shear elastic modulus (G′) starts to rise. At this stage, the cement mix is becoming progressively thicker and is less able to flow. In both cases, an exponential increase in storage shear elastic modulus (G′) was observed. In the case of ProRoot MTA the increase in G′ was not as consistent as in RS+ and was composed from at least two different regions, indicating more complex solidification. The final values of G′ were 1.77 × 10^8^ and 2.48 × 10^8^ Pa for ProRoot MTA and RS+, respectively, indicating higher stiffness of ProRoot MTA. Extrapolating the slope of the maximum increase in G′ provided a rough estimation of the time at which the transition from viscoelastic to elastic material occurred and could be considered as the time of solidification (setting), i.e., as a result of more comprehensive particle interlocking [33]. This transition occurred more than two times faster in the case of RS+ compared to ProRoot MTA (60 versus 120 min). The difference in attained higher stiffness for the ProRoot MTA could be ascribed to the material being still largely unreacted after 48 h of hydration and also consisting of a much larger PSD, providing a stiffer matrix connected with the thin film of hydrated products.

## 4. Discussion

In this study, differences in compositional and morphological characteristics, in hydration and phase transformation kinetics and rheological behaviour between ProRoot MTA and a novel RS+ were detected. Therefore, the null hypothesis was rejected. RS+ is considered a next generation synthetic HCSC with a refined physio-chemical formulation for improved handling, setting, and biological properties, while ProRoot MTA is a benchmark MTA material extensively studied and used clinically.

The morphologies and particle size distributions (PSDs) of both HCSCs were different. RS+ had a narrower mono-modal PSD with an order of magnitude-finer *d_mean_* value compared to ProRoot MTA, i.e., 1 µm versus 15 µm (Figure 3). ProRoot MTA particles, in addition to being larger, also resemble sharp edges and vertices (Figure 2), which could be related to the milling of clinker in Portland cement production [43]. RS+ powder, on the other hand, is substantially finer since it is produced synthetically, with micron-sized cuboid particles covered with finer, spherical-shaped particles (Figure 2b). Further distinctions between the HCSC powders were revealed with SEM-EDA mapping and XRD. The former confirmed a more complex composition of ProRoot MTA (Figure 4). Bismuth was detected, and in addition to calcium and silicon, sulphur, aluminium, and iron were detected as well. Bismuth oxide, used as the main radioopacifier, has been reported to be cytotoxic, negatively affecting cell proliferation [23]; thus, the newer HCSCs primarily use other biocompatible radiopaque materials, for example, biomedical grade zirconia [1], as is the case for RS+ as well. The XRD analysis confirmed the much more complex and diverse phase composition of ProRoot MTA (Figure 5) compared to RS+ (Figure 6a). In the case of the latter, homogeneously distributed calcium, silicon, and zirconium were detected. The variation in the concentration of calcium could be attributed to the distribution or agglomeration of other calcium-containing compounds present in the RS+ formulation, such as amorphous calcium silicate and/or bentonite.

The kinetics of surface hydration for both powders when mixed with water were analogous and comparable, as evidenced by the evolution of the time-dependent ATR FT-IR spectra (Figure 6 and Figure 7). Hydration reaction kinetics occur on the surface of the particles. The difference in PSDs between both HCSCs is reflected in their Raman spectra, where the RS+ paste is characterised by much broader, weaker, and less distinct bands in comparison with ProRoot MTA. This difference is typically observed in amorphous materials with short-range ordering. Moreover, strong carbonate bands arising from surface carbonation observed in the RS+ spectra but were absent in the case of ProRoot MTA, which might also be related to its fine-grained matrix, resulting in greater surface area and thus accelerating the atmospheric CO_2_ mineralisation with portlandite formed during the setting reaction.

The differences in chemical composition, an order of magnitude-finer PSD of RS+ coupled with the more uniform particulate morphology and addition of plasticizer (bentonite), proved pivotal in dictating the setting dynamics, namely in the ability to exert predictable viscous flow during working time (Figure 10a) and more than two times faster viscoelastic-elastic transition (Figure 10b), reflecting improved handling and setting behaviour. Bentonite is known for its ability to adsorb significant amounts of ions affecting the bentonite-containing slurries [44], high swelling capacity, and the formation of a gel-like structure with yield characteristics and viscoelastic properties at relatively low phyllosilicate clay concentrations [45,46]. Longer setting times for ProRoot MTA and the inhomogeneous sandy-like paste state are in agreement with the previous studies [16,26,27], in which even longer times of up to 300 min were reported to achieve a higher shear modulus [27].

RS+ could be completely transformed into hydration products, i.e., calcium silicate hydrate and calcium hydroxide, within 48 h, while hydration products were not yet detected by XRD in ProRoot MTA. In the case of the latter, the hydration was obviously bound to particulate surface thin film connecting the matrix of the unreacted particles, indicating that the hydration reactions were ongoing, albeit at a slower rate. The formed surface reaction film was responsible for an immediate and continuous increase in G′ for ProRoot MTA (Figure 10b), indicating ongoing solidification, in accordance with the previous findings [26,27].

Difficult handling is one of the main drawbacks of conventional MTA-based HCSCs. The inability to exert flow and slow hydration bound to the surface might be the reasons for the difficult adaptation of MTA materials in the root canal system or other defects [47]. An improved material’s rheological behaviour could represent a clinical advantage. A predictable viscous flow provides easier handling and material adaptation, while higher viscosity decreases the probability of material dislodgement during its application, such as periapical extrusion [47]. On the other hand, faster setting kinetics present a high clinical potential to reduce the impact of contamination with blood and other biological fluids, which affects the performance of HCSCs [48], also allowing for final coronal restoration in the same clinical session. Further, a shortened setting time could potentially decrease the risk of cement wash-out by blood flow in procedures, such as root end surgery, which require short setting times of about 3–10 min [5]. While the bioactivity of ProRoot MTA has previously been confirmed, showing the capacity to promote cell proliferation and differentiation [9,49] and mineral forming ability from phosphate-containing saliva, the bioactivity of RS+ should be further studied.

Because of the favourable rheological and faster setting kinetics, synthetic, finer grained HCSCs, such as RS+, represent a viable option as an alternative to conventional MTA-based materials for endodontic treatments. However, the promising in vitro material performance (physio-chemical properties) of RS+ in terms of the cement composition, fineness, setting kinetics, and rheological properties should be complemented by a controlled, long-term clinical study in the aggressive environment of the oral cavity.

## 5. Conclusions

Distinct differences were observed between the benchmark MTA-based material ProRoot MTA and RS+, a next generation HCSC with a refined physio-chemical formulation. RS+ had a narrower monomodal PSD distribution with an order of magnitude-finer *d_mean_* value compared to ProRoot MTA. SEM-EDA mapping confirmed the presence of homogeneously distributed calcium, silicon, and zirconium in the RS+, while in ProRoot MTA, in addition to calcium, silicon and bismuth, sulphur, aluminium, and iron were detected as well. The XRD analysis confirmed the much more diverse phase composition of ProRoot MTA.

While the kinetics of surface hydration of both HCSCs were comparable, it is concluded that the much finer biocompatible particulate formulation of RS+ proved pivotal in its ability to exert predictable viscous flow during the working time and more than two times faster viscoelastic-elastic transition, reflecting in improved handling and setting behaviour. Finally, RS+ could be completely transformed into hydration products within 48-h, while hydration products were not yet detected by XRD in ProRoot MTA since it bound to a surface thin film connected to the matrix of the unreacted particles.

## Figures and Tables

**Figure 1 materials-16-03174-f001:**
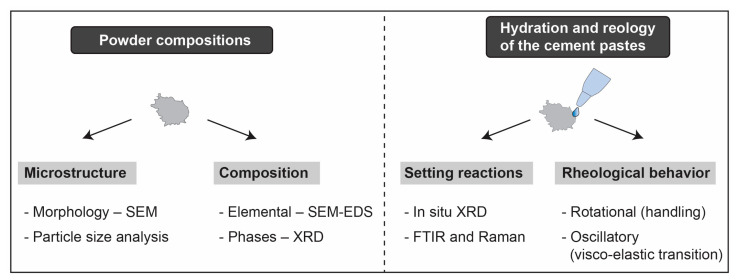
Experimental flowchart.

**Figure 2 materials-16-03174-f002:**
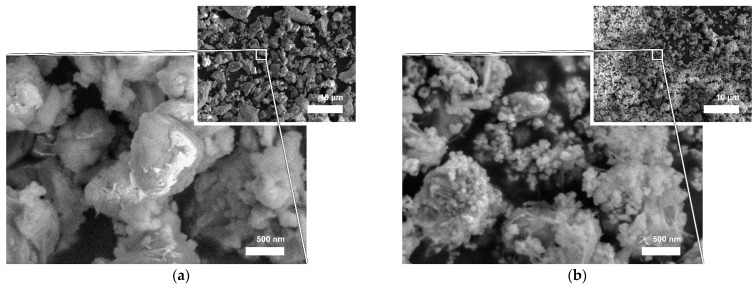
SEM micrographs showing the powder morphology, agglomeration state, and particle size: (**a**) ProRoot MTA and (**b**) RS+.

**Figure 3 materials-16-03174-f003:**
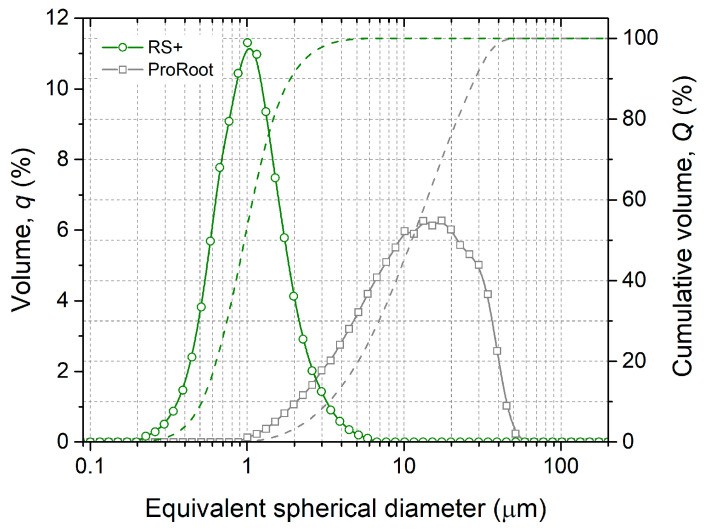
Particle size distributions of as-received hydraulic cements.

**Figure 4 materials-16-03174-f004:**
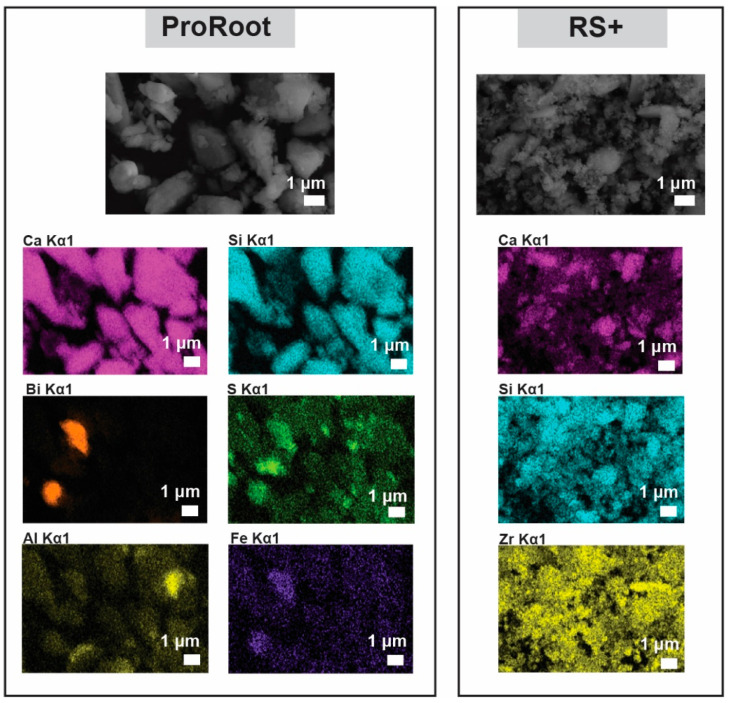
SEM-EDS mapping of the as-received powders showing elemental distribution.

**Figure 5 materials-16-03174-f005:**
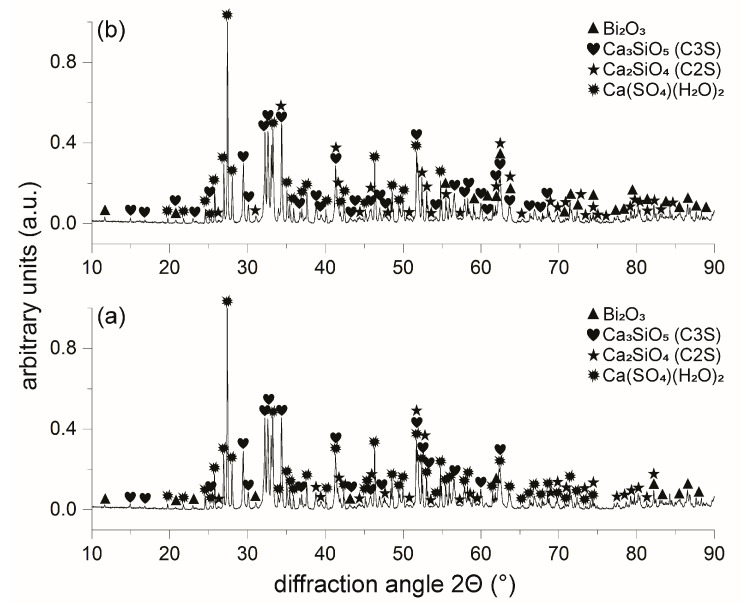
XRD diffractograms of the ProRoot MTA: (**a**) as received; and (**b**) after 48-h hydration.

**Figure 6 materials-16-03174-f006:**
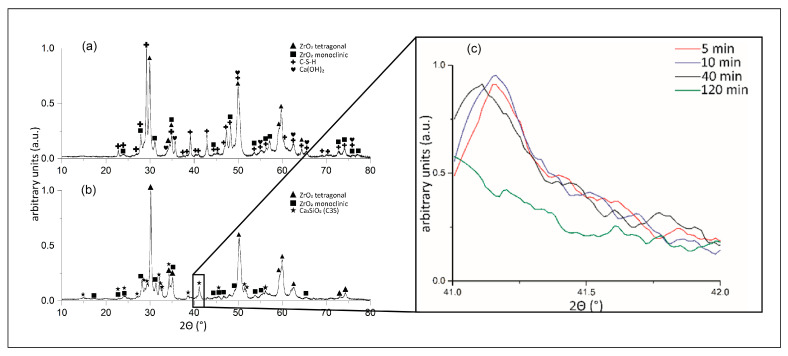
XRD diffractograms of the RS+ CSH cement: (**a**) as received; and (**b**) after hydration. (**c**) In situ XRD monitoring of the hydration kinetics in the 41–41.5 2θ range.

**Figure 7 materials-16-03174-f007:**
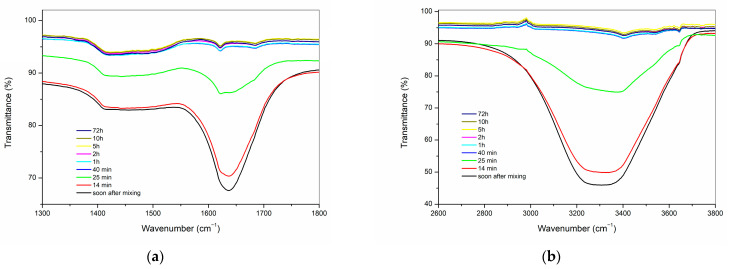
ATR-FTIR spectra of the ProRoot MTA paste at selected time intervals following mixing shown in the regions (**a**) 1300–1800 cm^−1^; (**b**) 2600–3800 cm^−1^.

**Figure 8 materials-16-03174-f008:**
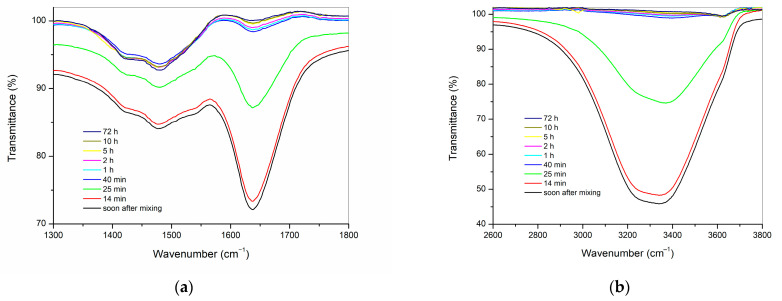
ATR-FTIR spectra of the RS+ paste at selected time intervals following mixing shown in the regions of: (**a**) 1300–1800 cm^−1^; and (**b**) 2600–3800 cm^−1^.

**Figure 9 materials-16-03174-f009:**
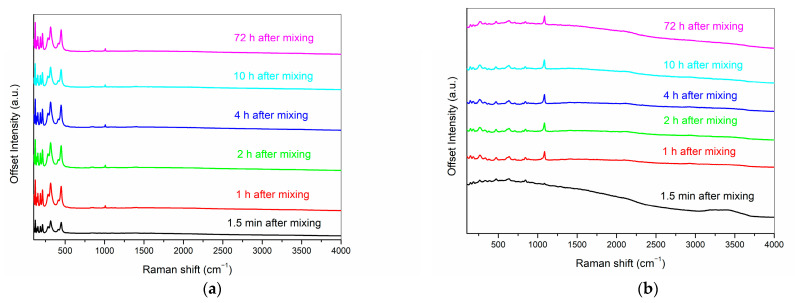
Raman spectra of the smeared ProRoot MTA (**a**) and RS+ (**b**) samples at selected time intervals following mixing. Individual spectra have been vertically offset for clarity.

**Figure 10 materials-16-03174-f010:**
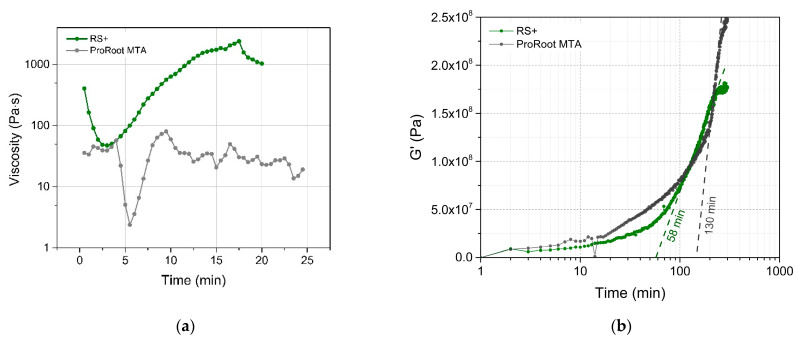
(**a**) Viscosity of RS+ and ProRoot MTA plotted as a function of time at the fixed shear rate; (**b**) storage modulus of RS+ and ProRoot MTA plotted as a function of time.

## Data Availability

Not applicable.

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
