# Peer review of "Rheological Properties and Setting Kinetics of Bioceramic Hydraulic Cements: ProRoot MTA versus RS+"

_materials, 2023, doi:10.3390/ma16083174_

Round 1

Reviewer 1 Report

The paper "Rheological Properties and Setting Kinetics of Bioceramic Hydraulic Cements: ProRoot MTA versus RS+" is well written and presents a relatively interesting topic for the literature in the area, however some adjustments are necessary:

(a) In the abstract, authors must insert the full name of all the microstructural techniques used, such as XRD, SEM and others... In addition, more quantitative results found must be added;

(b) In the introduction, the motivation of this research as well as the gap it really fills should be highlighted, in addition, the authors should delve into some discussions on the subject, some papers may be relevant to be considered, such as: 10.1016/ j.cscm.2022.e01271; 10.1021/ie980804b; 10.1016/j.biomaterials.2006.05.043.

(c) An experimental flowchart can be added in the methodology section, as well as a more detailed and in-depth description of the sample preparation techniques for the characterization analyzes used;

(d) The discussions presented could be integrated into the presentation of the results, the authors should better explain the phenomena based on comparisons in other studies of the international literature on the subject;

(e) The conclusion should be complemented with other more relevant aspects and findings that should be effectively highlighted by the authors.

None. 

Reviewer 2 Report

This is a well conducted study which demonstrated some improved setting behabior and rheological property of a synthetic hydraulic calcium silicate material (RS+) compared with ProRoot MTA. This reviewer has some minor concerns as follows:

1. Abstract

- Please spell out abbreviations (XRD, FTIR and SEM-EDS).

- Please include the conclusion of this study.

2. Introduction

Overall well structured, it provides most of the information necessary to understand the scientific background, the knowledge gap and the objectives of the study. However, to facilitate readers' understanding, this reviewer would recommend adding more information on the reported property of the newly developed material (RS+). What is the rationale for choosing this material among various synthetic HCSCs? 

3. Materials and Methods

- The scientific methodology used is in general described in a clear and exhaustive manner. 

Please spell out abbreviations (XRD and FTIR) at their first appearnce.

- L. 117: Please correct "weight".

- L. 127-8: Please provide manufacturer's information (including City, Country) for "FEI Helios NanoLab focused ion beam (FIB)-SEM".

- L. 143: Please provide manufacturer's information for "Malvern Panalytical Empyrean X-ray diffractometer"

4. Results

- All the results are presented as representative findings. How many times was each analysis repeated? Was there any possibility of conducting statistical analysis?

5. DISCUSSION

- The discussion of the results is on the whole well articulated. However, limitations of the study should be emphasized more.

- Please explain by which mechanisms the addition of bentonite changes the setting kinetics of RS+, with siting some references.

Round 2

Reviewer 1 Report

All corrections were performed by the authors.

All corrections were performed by the authors.